# 3DGraphLLM: Combining Semantic Graphs and Large Language Models for 3D Referred Object Grounding

## Abstract

A 3D scene graph represents a compact scene model, storing information about the objects and the semantic relationships between them, making its use promising for robotic tasks. When interacting with a user, an embodied intelligent agent should be capable of responding to various queries about the scene formulated in natural language. Large Language Models (LLMs) are beneficial solutions for user-robot interaction due to their natural language understanding and reasoning abilities. Recent methods for creating learnable representations of 3D scenes have demonstrated the potential to improve the quality of LLMs responses by adapting to the 3D world. However, the existing methods do not explicitly utilize information about the semantic relationships between objects, limiting themselves to information about their coordinates. In this work, we propose a method 3DGraphLLM for constructing a learnable representation of a 3D scene graph. The learnable representation is used as input for LLMs to perform 3D vision-language tasks. In our experiments on popular ScanRefer, RIORefer, Multi3DRefer, ScanQA, Sqa3D, and Scan2cap datasets, we demonstrate the advantage of this approach over baseline methods that do not use information about the semantic relationships between objects.

## 1 Introduction

In this paper, we consider scene understanding in the context of solving 3D vision-language problems: 3D referred object grounding task, 3D dense scene captioning and 3D visual question answering. The 3D referred object grounding task involves identifying a region in a 3D scene that corresponds to a complex natural language query. This query may describe object properties (e.g., color, size) and spatial relationships (e.g., a mug on a table). A common approach to solving this problem is to assume that one is given a 3D reconstruction of the scene (e.g., a point cloud, mesh, or NeRF). The goal is to predict the bounding boxes of the region that matches the query. The goal of the dense scene captioning task is to describe the selected object. Finally, the goal of the 3D visual question answering task is to generate text answer to various questions about the properties of the scene. It seems promising to explicitly use a three-dimensional scene graph to solve these tasks.

The 3D scene graph not only allows storing multimodal information about individual objects within a scene but also captures the semantic relationships (Wang et al., 2023b; Koch et al., 2024) and hierarchical organization between them (Werby et al., 2024; Honerkamp et al., 2024). Additionally, the graph scene representation enables real-time updates for dynamic environments (Rosinol et al., 2021; Özsoy et al., 2023), and supports the application of graph algorithms for tasks such as navigation (Zhou et al., 2023b; He & Zhou, 2024; Honerkamp et al., 2024) or object search based on textual queries (Feng et al., 2021; Chang et al., 2023; Werby et al., 2024; Gu et al., 2024).

The solving of 3D vision-language tasks (Chen et al., 2020; 2021; Azuma et al., 2022) is crucial for embodied intelligent agents. To interact with the user, an intelligent agent must be able to describe the environment and answer questions about its properties using natural language. Large language models (LLMs) are particularly well-suited for this task, as their advanced capabilities in natural language understanding and common-sense reasoning make them highly effective in interpreting and

Figure 1: Proposed *3DGraphLLM* approach leverages 3D semantic scene graph learnable representation supplied as input to an LLM to perform various 3D vision-language tasks.

matching user queries to objects in a scene (Hong et al., 2023b; Wang et al., 2024; Gu et al., 2024). Using LLMs makes it easier to adapt the method to new categories of objects and relationships found in referring expressions. LLMs can also handle complex queries that don't explicitly mention the class name, but instead describe its function (e.g. "somewhere to sit").

The 3D scene description input for LLMs can be represented either as text (Gu et al., 2024; Linok et al., 2024; Werby et al., 2024; Honerkamp et al., 2024; Yang et al., 2024; Yuan et al., 2024), or through learnable representations (Hong et al., 2023b; Chen et al., 2023; Huang et al., 2023; Chen et al., 2024; Cheng et al., 2024), which encode objects and their relationships using significantly fewer tokens and their corresponding embeddings than a textual description of the scene. These learnable representations enhance the performance of the LLM in generating responses to user queries, while also improving response accuracy through adaptation to 3D scenes. However, current methods (Hong et al., 2023b; Chen et al., 2023; Huang et al., 2023; Chen et al., 2024) for 3D vision-language tasks using LLM and learnable 3D scene representations fail to leverage semantic relationships between objects, relying solely on their spatial coordinates.

In this paper, we propose a novel learnable representation of a 3D scene graph called 3DGraphLLM, designed for use as input to a LLM (see Figure 1). This representation consists of a list of learnable embeddings for objects within the scene, where each object is represented by a subgraph containing the object itself along with several of its nearest neighbors. These object subgraphs are provided to the LLM as a sequence of triplets *(object1, relation, object2)*. Semantic relations between objects are embedded using features derived from the semantic edges of the graph, which is generated using state-of-the-art methods for 3D semantic graph generation such as VL-SAT (Wang et al., 2023b). Our experiments demonstrate that incorporating semantic relationships between objects significantly improves the accuracy of LLM responses for 3D vision-language tasks, outperforming baseline approaches for creating learnable scene representations.

**To summarize**, our contributions are as follows:

- We introduce 3DGraphLLM, the first method to create a learnable 3D scene graph representation for LLMs, enabling the mapping of semantic relationships between objects in the scene to LLM's token embedding space.

- We propose an algorithm that produces a flat sequence of graph embedding tokens using k-nearest neighbor selection with a minimum distance filter between objects, optimizing inference speed by reducing the number of tokens required to describe the scene.

- 3DGraphLLM shows state-of-the-art results for the 3D referred object grounding task on the Multi3DRefer (Zhang et al., 2023) (+5.8% F1@0.5) and ScanRefer (Chen et al., 2020) (+4.4% Acc@0.5) benchmarks and also for the 3D scene captioning on the Scan2Cap dataset Chen et al. (2021) (CIDEr@0.5 +5.8%).

The code for training and inference of 3DGraphLLM will be made publicly available, with all training and validation performed on open datasets.

## 2 RELATED WORKS

**Scene Graphs.** The concept of a scene graph was initially developed for 2D images, providing a structured representation of a scene's semantics by incorporating relationships between the semantic elements (Johnson et al., 2015). In the context of images, scene graphs have proven effective for tasks such as content-based image retrieval (Johnson et al., 2015; Pei et al., 2023), 2D referring expression comprehension (Yang et al., 2019a; Shi et al., 2023; Han et al., 2024), image caption (Yang et al., 2019b; Phueaksri et al., 2023), image generation (Johnson et al., 2018; Farshad et al., 2023).

In 3D scenes, a scene graph is commonly used to address robotics challenges such as planning (Werby et al., 2024; Honerkamp et al., 2024), object grounding for navigation (Werby et al., 2024; Gu et al., 2024; Linok et al., 2024; Honerkamp et al., 2024) and manipulation (Honerkamp et al., 2024), as well as scene generation (Zhai et al., 2024; Gao et al., 2024).

Our approach is part of a class of methods that utilize an implicit representation of the scene graph, such as OVSG (Chang et al., 2023), which frames the problem of 3D object grounding as subgraph retrieval. 3DGraphQA (Wu et al., 2024) proposes to use the bilinear graph neural network for feature fusion between scene and question graphs for question answering task. Feng et al. (2021) build a graph based on a text query, which is used to refine the visual graph in order to select from its vertices the one that best fits the description. However, the application scope of this method is limited to specific tasks as 3D referred object grounding with one referred object or question answering. In contrast, we propose a more versatile method capable of solving various 3D vision-language tasks.

**3D Language Scene Understanding**. 3D scene understanding is a complex computer vision task that involves identifying the semantic, physical, and functional properties of objects, as well as their mutual relations. One of the goals of 3D scene understanding is to develop methods capable of responding to natural language queries about the scene. The queries may correspond to different visual-language tasks such as 3D referred object grounding (Chen et al., 2020; Zhang et al., 2023; Miyanishi et al., 2024), question answering (Azuma et al., 2022), and dense scene captioning (Chen et al., 2021). Recent approaches address these queries by reconstructing the scene as a 3D mesh (Peng et al., 2023) or point cloud (Zhao et al., 2021; Chen et al., 2022; Zhu et al., 2023), often enhanced with instance segmentation (Zhu et al., 2023).

The emergence of transformer models (Vaswani, 2017) has enabled the development of neural network models that create a learnable representation of a scene for answering various language queries. MultiCLIP (Delitzas et al., 2023) proposes to align 3D scene representation with text queries and multi-view 2D CLIP (Radford et al., 2021) embeddings to improve the quality of question answering. 3DVG-Transformer (Zhao et al., 2021) and Vil3DRef (Chen et al., 2022) methods introduce modules for modeling spatial relationships between objects to improve the quality of object grounding. 3D-VisTA (Zhu et al., 2023) presents a transformer model for aligning 3D object and text representations, coupled with an unsupervised pre-training scheme to solve various 3D vision-text problems using specialized task-specific heads. However, these fully supervised approaches face challenges in generalizing to new tasks and domains. In contrast, leveraging large language models (LLMs) for scene understanding enhances generalization capabilities and taps into the extensive knowledge LLMs contain about the physical world (Hong et al., 2023b).

**Large Language Models for Scene Understanding.** Large language models (LLMs) offer several advantages for scene understanding, notably enhancing the ability to address complex queries that require common knowledge. LLMs can serve as agents that decompose user queries into elementary tasks, which can then be addressed by other methods (Yang et al., 2024; Yuan et al., 2024). Additionally, LLMs can act as an interface for reasoning by processing textual descriptions of the scene as input (Linok et al., 2024; Gu et al., 2024). BBQ (Linok et al., 2024) and ConceptGraphs (Gu et al., 2024) demonstrate that using a text-based graph representation with an LLM interface significantly improves the quality of object retrieval compared to using CLIP features of objects. HOV-SG (Werby et al., 2024) construct a hierarchical graph consisting of objects, rooms, and floors, and demonstrate the effectiveness of such a representation for the task of object grounding given a query containing object location hints. The authors of the MOMA (Honerkamp et al., 2024) method propose using a hierarchical scene graph together with a navigational Voronoi graph as input to LLM to predict a high-level policy for object search for navigation and manipulation. However, using text to describe an object in a scene graph inevitably leads to the loss of some of the information contained in its RGB point cloud. Additionally, in the case of using a text graph, several hundred tokens may be

required to describe one object (its semantic class, pose), which will significantly slow down LLM inference in the case of a large number of objects in the scene.

Recent advancements have successfully integrated point cloud data into LLMs by employing pre-trained point cloud encoders and training adapters to align the resulting representations with the LLM embedding space. 3D-LLM (Hong et al., 2023a) aggregates 3D point cloud features from a sequence of 2D images and then solves the grounding problem as a prediction of a sequence of location tokens added to the LLM dictionary. Chat3D-v2 (Huang et al., 2023) generates a 3D feature for each object in the scene and then treats the grounding problem as an object selection problem. LLA3D (Chen et al., 2023) proposes to use a set of trainable fixed-length query tokens obtained by interacting potential visual cues, text cues, and object point cloud features in a transformer model. Grounded 3D-LLM (Chen et al., 2024) uses referent tokens to decode object masks in point clouds. Additionally, research has demonstrated that incorporating spatial information, such as object coordinates (Huang et al., 2023) or depth maps (Cheng et al., 2024), enhances the accuracy of responses to user queries.

Despite recent advances, existing methods do not fully leverage the rich semantic information in object relationships. In this paper, we introduce 3DGraphLLM, a method that demonstrates the effectiveness of utilizing semantic relationships between objects to enhance performance across various scene understanding tasks.

## 3 METHOD

Our approach uses a set of point clouds of scene objects as input. The objects' point clouds can be obtained either from ground-truth annotations or through state-of-the-art point cloud instance segmentation methods. These point clouds are used to extract scene graph features (see Section 3.1). A scene graph consists of nodes representing the objects and edges corresponding to semantic relationships between them. To convert the scene graph into a token sequence, we represent each object by an identifier, followed by a subgraph comprising the object's $k$ nearest neighbors. The relationships between an object and its neighbors are encoded as triplets $(object_i, relation_{ij}, object_j)$. The scheme of the 3DGraphLLM approach is shown in Figure 2. For more details on the scene graph representation, refer to Section 3.2. Our training process is two-stage. First, we pre-train the model on a dataset for various 3D scene understanding tasks using ground-truth instance segmentation. Next, we fine-tune 3DGraphLLM with predicted instance segmentation of scene point clouds, considering a scenario where ground-truth segmentation is unavailable (see Section 3.3).

### 3.1 MODEL ARCHITECTURE

The model architecture includes pre-trained encoders for 3D point clouds and their semantic relationships, alongside a pre-trained LLM. We train projection layers to map the extracted object features and their relationships into the LLM's token embedding space. Following the approach of Chat-Scene (Huang et al., 2024), we introduce additional object identifier tokens $\{< \texttt{OBJ}i >\}_{i=1}^{n}$ into the LLM's vocabulary. Here and throughout, we use $n$ to denote the number of objects in the scene. These learned identifiers, along with the features from object subgraphs composed of nearest neighbors for each object, are used to create a flat representation of the scene graph, which is then fed into the LLM.

**Object Proposals.** We use point clouds of objects in the scene as vertices in the scene graph $G$. In our experiments, we evaluate 3DGraphLLM in various modes, including ground-truth scene segmentation and instance segmentation using state-of-the-art neural network methods like Mask3D (Schult et al., 2023) and OneFormer3D (Kolodiazhnyi et al., 2024). Thus, the set $V$ of vertices of the graph consists of $n$ point clouds $\{P_i\}_{i=1}^{n}$, where $P_i \in \mathbb{R}^{m_i \times 6}$. Here, $m_i$ is the number of points in the $i$-th object proposal of instance segmentation of scene point cloud, and 6 dimensions of each point correspond to its 3D coordinates and RGB color.

**Object Identifiers.** Following the approach in Chat3D-Scene, we add a set of learnable identifier tokens $\{< \texttt{OBJ}i >\}_{i=1}^{n}$ to the LLM's vocabulary for object identification. These tokens allow the model to identify objects in the scene by simply predicting the corresponding object identifier token. In our experiments, we assume a maximum of 200 objects per scene.

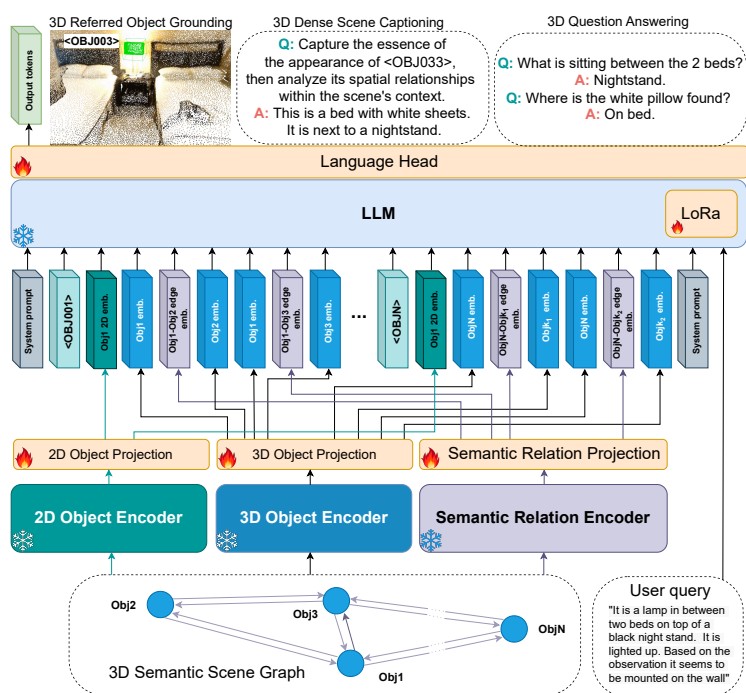

Figure 2: The overall architecture of our approach. 3DGraphLLM leverages pre-trained encoders for 3D object point clouds and semantic relationships between objects. We introduce trainable layers to map the extracted graph node and edge features into the token embedding space of a pre-trained LLM. The scene graph is flattened for input into the LLM, with each object represented by a subgraph of its $k$ nearest neighbors. To further adapt the LLM to 3D vision-language tasks, we add new object tokens to the LLM's vocabulary and fine-tune it using LoRa.

**2D Object Encoder.** The results of Chat-Scene demonstrate that adding aggregated 2D DINOv2(Oquab et al., 2023) features increase the LLM performance on 3D vision-language tasks. Therefore, we add DINOv2 $Z_i^{2d} \in \mathbb{R}^{1 \times 1024}$ features as an additional token describing the object subgraph. DINOv2 object features are obtained by aggregating features from the masked multi-view images where masks come from the projection of the object's 3D point cloud..

**3D Object Encoder.** We extract vertex features using a pre-trained Uni3D (Zhou et al., 2023a) encoder, which generates point cloud features aligned with their textual descriptions. Since this model is pre-trained on a large dataset, it enables us to produce high-quality graph vertex embeddings across various data domains. For each object point cloud $P_i$, we extract Uni3D feature $Z_i^{v_p} \in \mathbb{R}^{1 \times 1024}$.

**Edge Feature Encoder.** One challenge in generating features for semantic relationships between objects is that most methods for 3D semantic scene graph generation are trained on 3RScan scenes (Wald et al., 2019), while visual grounding tasks are typically tested on ScanNet scenes (Dai et al., 2017). Although both datasets belong to the indoor scene domain, existing methods struggle with performance in cross-domain testing, resulting in a drop in accuracy for the grounding task (Miyanishi et al., 2024).

To extract semantic relationships between objects, we use VL-SAT (Wang et al., 2023b), a method for generating 3D semantic scene graphs from point clouds. One of its key advantages is that it only requires 3D point cloud coordinates as input during prediction, while leveraging knowledge transfer from the pre-trained CLIP model (Radford et al., 2021). This allows the method to perform well when applied to new scene domains (Wang et al., 2023b), as confirmed by our experiments (see Section 4.3 and Tables 3 and 4). For each pair of point clouds $P_i$ and $P_j$, we generate a latent feature representing their relationship $Z_{ij}^e \in \mathbb{R}^{1 \times 512}$, which corresponds to VL-SAT graph neural network feature before the classification head assigning semantic categories to the graph edges.

Table 1: Example of prompt for the language model containing scene graph.

| | |
|---|---|
| System: | A chat between a curious user and an artificial intelligence assistant. |
| | The assistant gives helpful, detailed, and polite answers to the user's questions. The conversation centers around an in- |
| | door scene: [<OBJ001> $F_1^{2d}, F_1^v, F_{12}^e, F_2^v F_1^v, F_{14}^e, F_4^v$ ...<OBJN> $F_N^{2d}, F_N^v, F_{Nk_1}^e, F_{k_1}^v F_N^v, F_{Nk_2}^e, F_{k_2}^v$] |
| User: | According to the given description, *there are brown wooden cabinets*, |
| | *placed on the side of the kitchen*, please provide the ID of the object that closely matches this description. |
| Assistant: | <OBJ001>. |

While VL-SAT predicts a fixed set of relationships between objects, these relationships are not mutually exclusive (e.g., "larger" and "close"). Therefore, we use latent features to capture possible combinations of these semantic relationships.

**2D/3D object, and semantic relation projection.** To adapt the extracted features for the language model, we use three trainable projection modules: the 2D Object Projection $f_{2d}(\cdot)$, which maps the 2D image features of objects, the 3D Object Projection $f_v(\cdot)$, which maps the point cloud features of objects, and the Semantic Relation Projection $f_e(\cdot)$, which maps the features of semantic relationships between objects. Therefore, for the $i$-th object, the 2D and 3D object features are projected to token embeddings $F_i^v$ and $F_i^{2d}$ respectively. For the pair of $i$-th and $j$-th objects, the semantic relation feature is projected to token embedding $F_{ij}^e$:

$$F_i^{2d} = f_v(Z_i^{2d}), F_i^v = f_v(Z_i^v), F_{ij}^e = f_e(Z_{ij}^e). \tag{1}$$

### 3.2 FLAT GRAPH REPRESENTATION

The scene graph is a complete graph because we can generate connections between all pairs of objects. However, such a graph contains $n \cdot (n-1)$ edges between objects, and using the complete graph as a sequence for the LLM would significantly increase the sequence length. However, intuitively, the most relevant relationships for answering user questions are those between an object and its nearest neighbors. Therefore, for each object, we consider a subgraph of its $k$ nearest neighbors. The relationships between objects are encoded using features extracted from point clouds $\{F_i^v\}_{i=1}^n$ and semantic relations features $\{F_{ij}^e, i \in \{1, ..., n\}, j \in \{1, ..., n\}\}$, represented as a triplet $(F_i^v, F_{ij}^e, F_j^v)$. When using the complete scene graph, the number of tokens required to describe the scene is $2 \cdot n + 3n \cdot (n-1)$. For 100 objects, which matches the number of objects in the Mask3D (Schult et al., 2023) instance segmentation, this totals 29900 tokens. By using a $k$-nearest neighbor subgraph, we reduce the token count to $n + 3n \cdot k$. As shown in Section 4.3 (see Figure 4), setting $k = 2$ improves accuracy in 3D visual-language tasks while reducing the number of tokens needed to describe a scene with 100 objects to 800.

**Prompt template.** Thus, we integrate the scene description as a sequence of object subgraphs into the prompt for LLM in the following way, similar to the integration of the list of objects and their embeddings in the Chat-Scene method (Huang et al., 2024). An example of a prompt for LLM containing a system prompt, a scene description in the form of an object identifier and an object subgraph, a user request, and an LLM assistant response is given in Table 1. The sequence describing an object $i$ starts with its identification token <OBJi>. Then there are $k$ triplets $\{(F_i^v, F_{ij_k}^e, F_{j_k}^v)\}_{j_k=1}^k$ describing the relationship between the object and its $k$ nearest neighbors.

### 3.3 TRAINING STRATEGY

Following the strategy used in Chat-Scene(Huang et al., 2024), we implement a training approach that involves simultaneously training the projection layers and the language model. We also conduct joint training for various tasks, including visual grounding (ScanRefer (Chen et al., 2020) and Multi3DRefer (Zhang et al., 2023)), 3D scene description (Scan2Cap (Chen et al., 2021)), and 3D visual question answering (ScanQA (Azuma et al., 2022) and SQA3D (Ma et al., 2022)). This adaptation of the tasks is designed for user-assistant interactions, as proposed by the authors of Chat-Scene. During training, we aim to optimize the trainable parameters $\theta$ of both the language model and the projection layers to minimize the negative log-likelihood of the target response $s^{\text{res}}$ compared to the response predicted by the model. We use the loss function from the Chat-Scene

method, adapting it to fit our proposed graph representation of the scene given the input prefix sequence $s^{\text{prefix}}$ containing system and user prompts:

$$L(\theta) = -\sum_{i=1}^{\ell} \log P(s_i^{\text{res}} | s_{[1,\dots,i-1]}^{\text{res}}, s^{\text{prefix}}), \qquad (2)$$

where $\ell$ is the length of the token sequence in the LLM response, $s_{[1,\dots,i-1]}^{\text{res}}$ is the sequence generated up to the $i$-th token. The trainable parameters $\theta$ include the parameters of 3D Object Projection and Semantic Relation Projection Layers, added object identifier token embeddings and the language model.

We use the encoder for semantic relationships between objects pre-trained using ground-truth (GT) point cloud scene segmentation data (Wang et al., 2023b). Since the predicted point cloud segmentation typically contains more noise than the GT segmentation, we anticipate that the edge features derived from the GT segmentation will be of higher quality than those from the neural network instance segmentation. To address this problem, we employ a two-stage training strategy for 3DGraphLLM. First, we pre-train the projection layers and the language model on the GT instance segmentation data to achieve effective projections of the semantic embeddings of relations and objects into the language model's embedding space. Then, we fine-tune 3DGraphLLM using the noisy data from the neural network segmentation.

## 4 EXPERIMENTS

**Datasets.** We conduct experiments using the ScanNet (Dai et al., 2017) and 3RScan (Wald et al., 2019) scene datasets. For training 3DGraphLLM on ScanNet scenes (Dai et al., 2017), we utilize data from five 3D vision-language benchmarks: visual grounding tasks (ScanRefer (Chen et al., 2020), Multi3DRefer (Zhang et al., 2023)), scene description (Scan2Cap (Chen et al., 2021)), and 3D visual question answering (ScanQA (Azuma et al., 2022), SQA3D (Ma et al., 2022)). Each of these datasets follows a standard split into training and validation sets, corresponding to 1201 training scans and 312 validation scans from ScanNet. Additionally, we include the RioRefer dataset (Miyanishi et al., 2024), which provides referring expressions for objects in 3RScan scenes (Wald et al., 2019) splitting into standard training and validation sets (1175 training scans and 157 validation scans). Since our method primarily targets visual grounding tasks, the majority of validation experiments are performed on the ScanRefer, Multi3DRefer, and RioRefer datasets.

**Implementation details.** The projection layers for 3D object features and their semantic relations are three-layer MLPs. In our experiments, we use LLAMA3-8B-Instruct (AI@Meta, 2024), a state-of-the-art large language model, as well as Vicuna-1.5-7B (Zheng et al., 2023) for ablation. For fine-tuning the language model, we apply LoRA (Hu et al., 2021) with a rank of 16. We use a batch size of 8 and train 3DGraphLLM for 3 epochs with an initial learning rate of 0.00002, following a cosine annealing schedule. Training is performed on a server equipped with an NVIDIA A100 GPU, and the entire training process takes approximately 36 hours. In our experiments, we select $k = 2$ nearest neighbors to construct object subgraphs and, in the case of using Mask3D (Schult et al., 2023) instance scene point cloud segmentation, we use a NMS filter and a filter that ensures a minimum distance between nearest neighbors of 1 cm (see Section 4.3).

**Evaluation metrics.** For the visual grounding task on the ScanRefer (Chen et al., 2020) and RioRefer (Miyanishi et al., 2024) datasets, we use the standard metrics Acc@0.25 and Acc@0.5. A prediction is considered a true positive if the intersection-over-union (IoU) between the predicted object's 3D bounding box and the ground truth exceeds the thresholds of 0.25 and 0.5, respectively. The Multi3DRefer (Zhang et al., 2023) dataset contains queries that may refer to multiple objects. Therefore, we use the benchmark-standard F1 score at IoU thresholds of 0.25 and 0.5. During ablation experiments, we also assess the quality of object descriptions using the Scan2Cap (Chen et al., 2021) benchmark metrics CIDEr@0.5 and BLEU-4@0.5. For the visual question answering task, we follow the validation strategy from Chat3Dv2, applying CIDEr (Vedantam et al., 2015) and BLEU-4 (Papineni et al., 2002) metrics for ScanQA (Azuma et al., 2022), and exact match accuracy (EM) for SQA3D (Ma et al., 2022).

Table 2: Performance comparison of 3DGraphLLM with state-of-the-art approaches for 3D vision-language tasks. "Expert models" use specialized heads to deal with different 3D vision-language tasks. Our approach falls into the category of "LLM-based models" that consider different tasks as different user queries to a generative model. C denotes the CIDEr metric.

|  | Methods | ScanRefer | | Multi3DRefer | | ScanQA | | Sqa3D | Scan2Cap | |
|---|---|---|---|---|---|---|---|---|---|---|
|  |  | A@0.25↑ | A@0.5↑ | F1@0.25↑ | F1@0.5↑ | C↑ | B-4↑ | EM↑ | C@0.5↑ | B-4@0.5↑ |
| Expert models | ScanRefer (Chen et al., 2020) | 37.3 | 24.3 | - | - | - | - | - | - | - |
|  | MVT (Huang et al., 2022) | 40.8 | 33.3 | - | - | - | - | - | - | - |
|  | 3DVG-Trans (Zhao et al., 2021) | 45.9 | 34.5 | - | - | - | - | - | - | - |
|  | ViL3DRel (Chen et al., 2022) | 47.9 | 37.7 | - | - | - | - | - | - | - |
|  | M3DRef-CLIP (Zhang et al., 2023) | 51.9 | 44.7 | 42.8 | 38.4 | - | - | - | - | - |
|  | Scan2Cap (Chen et al., 2021) | - | - | - | - | - | - | - | 35.2 | 22.4 |
|  | ScanQA (Azuma et al., 2022) | - | - | - | - | 64.9 | 10.1 | - | - | - |
|  | Sqa3D (Ma et al., 2022) | - | - | - | - | - | - | 47.2 | - | - |
|  | 3D-VisTA (Zhu et al., 2023) | 50.6 | 45.8 | - | - | 72.9 | 13.1 | 48.5 | 66.9 | 34.0 |
|  | BUTD-DETR (Jain et al., 2022) | 52.2 | 39.8 | - | - | - | - | - | - | - |
|  | PQ3D (Zhu et al., 2025) | - | 51.2 | - | 50.1 | 87.8 | - | 47.1 | 80.3 | 36.0 |
| LLM-based models | ZSVG3D (Yuan et al., 2024) | 36.4 | 32.7 | - | - | - | - | - | - | - |
|  | 3D-LLM(Flamingo) (Hong et al., 2023a) | 21.2 | - | - | - | 59.2 | 7.2 | - | - | - |
|  | 3D-LLM(BLIP2-flant5) (Hong et al., 2023a) | 30.3 | - | - | - | 69.4 | 12.0 | - | - | - |
|  | Chat-3D v2 (Huang et al., 2023) | 35.9 | 30.4 | - | - | 77.1 | 7.3 | - | - | - |
|  | Scene-LLM (Fu et al., 2024) | - | - | - | - | 80.0 | 12.0 | 54.2 | - | - |
|  | LL3DA (Chen et al., 2023) | - | - | - | - | 76.8 | 13.5 | - | 65.2 | 36.8 |
|  | Grounded 3D-LLM (Chen et al., 2024) | 47.9 | 44.1 | 45.2 | 40.6 | 72.7 | 13.4 | - | 70.6 | 35.5 |
|  | Chat-Scene (Huang et al., 2024) | 55.5 | 50.2 | 57.1 | 52.4 | 87.7 | 14.3 | 54.6 | 77.1 | 36.3 |
|  | 3DGraphLLM Vicuna-1.5 (ours) | 57.0 | 51.3 | 60.1 | 55.4 | 87.6 | 12.1 | 53.1 | 81.2 | 36.3 |
|  | **3DGraphLLM LLAMA3-8B (ours)** | **60.2** | **54.6** | **63.0** | **58.2** | 83.1 | 12.5 | **55.2** | **82.9** | **37.8** |

User query:
This is a large twin sized bed. It is on the right side of the hotel room with a small pair of pants on it.

User query:
It is a black suitcase on the floor. It is sitting beside the mini fridge.

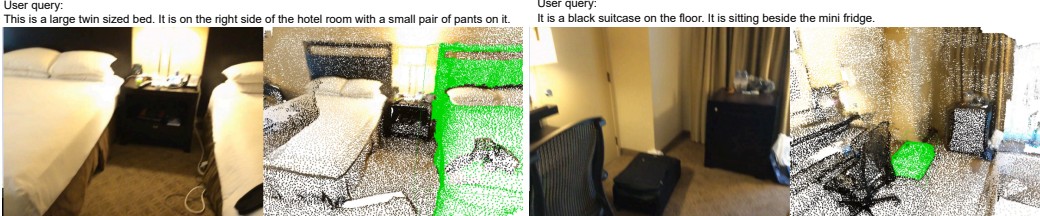

Figure 3: Qualitative examples of 3DGraphLLM performance on the ScanRefer dataset. For each query, we provide an RGB image from the ScanNet dataset showing the selected object, along with a visualization of the RGB point cloud. In the point cloud, green points indicate the points that 3DGraphLLM identified as corresponding to the object from the text query, while the green box highlights the ground truth (GT) box for the query.

## 4.1 EXPERIMENTAL RESULTS

**Comparison with state-of-the-art approaches.** As shown in Table 2, our method significantly outperforms baseline approaches that use LLMs on the two ScanNet 3D referred object grounding benchmarks, ScanRefer (Chen et al., 2020) and Multi3DRefer (Zhang et al., 2023), as wall on the Scene Captioning benchmark Scan2Cap (Chen et al., 2021). These results highlight the effectiveness of a learnable graph-based scene representation 3D vision-language tasks. It's worth noting that the performance of our method is comparable to state-of-the-art specialized models with separate heads for different language tasks, such as 3D-VisTA (Zhu et al., 2023), PQ3D (Zhu et al., 2025) and M3DRef-CLIP (Zhang et al., 2023). Notably, 3DGraphLLM demonstrates a clear advantage over PQ3D (Zhu et al., 2025) and M3DRef-CLIP (Zhang et al., 2023) on the Multi3DRefer dataset.

**Qualitative results.** Figure 3 shows the qualitative results of 3DGraphLLM on the ScanRefer dataset using Mask3D (Schult et al., 2023) instance scene segmentation. In the left part of the figure, 3DGraphLLM correctly identifies the bed on the right and leverages an additional spatial cue - pants that are lying on the bed. In the right part of the figure, 3DGraphLLM distinguishes the black suitcase next to the refrigerator, despite there being another suitcase farther away from the refrigerator in the scene.

## 4.2 ABLATION STUDIES. ROLE OF SEMANTIC RELATIONS AND TRAINING PIPELINE

To isolate the impact of using a scene graph representation, we conduct an experiment with different LLMs and training pipelines using Mask3D (Schult et al., 2023) instance segmentation. We train a

Table 3: Ablation study on semantic edges role and training pipeline. C denotes the CIDEr metric.

| Methods | Pre-train | Number of edges | ScanRefer Acc@0.5↑ | Multi3DRefer F1@0.5↑ | ScanQA C↑ | ScanQA B-4↑ | Sqa3D EM↑ | Scan2Cap C@0.5↑ | Scan2Cap B-4@0.5↑ |
|---|---|---|---|---|---|---|---|---|---|
| 3DGraphLLM-0 Vicuna1.5 | ✗ | 0 | 50.2 | 52.4 | 87.7 | 14.3 | 54.6 | 77.1 | 36.3 |
| 3DGraphLLM-2 Vicuna1.5 | ✗ | 2 | 50.1 | 52.7 | **92.2** | **15.5** | **54.7** | 80.4 | 36.9 |
| **3DGraphLLM-2** Vicuna1.5 | ✓ | **2** | **51.3** | **55.4** | 87.6 | 12.1 | 53.1 | **81.2** | **36.3** |
| 3DGraphLLM-0 LLAMA3-8B | ✗ | 0 | 52.0 | 55.1 | 84.0 | **15.8** | 53.8 | 80.0 | 37.5 |
| 3DGraphLLM-2 LLLAMA3-8B | ✗ | 2 | 54.3 | 57.3 | **87.4** | 14.9 | 54.5 | **85.6** | **39.6** |
| **3DGraphLLM-2** LLLAMA3-8B | ✓ | **2** | **54.6** | **58.2** | 83.1 | 12.5 | **55.2** | 82.9 | 37.8 |

Table 4: Ablation study on semantic edges role depending on quality of instance segmentation.

| Methods | Instance segmentation | Number of edges | Minimal distance, cm | ScanRefer Acc@0.25↑ | ScanRefer Acc@0.5↑ |
|---|---|---|---|---|---|
| 3DGraphLLM-0 | GT | 0 | - | 48.9 | 48.9 |
| **3DGraphLLM-2** | **GT** | **2** | **0** | **54.4**(+5.6%) | **54.4**(+5.6%) |
| 3DGraphLLM-0 | Mask3D | 0 | - | 46.0 | 34.2 |
| 3DGraphLLM-2 | Mask3D | 2 | 0 | 47.3(+1.3%) | 35.6(+1.4%) |
| 3DGraphLLM-2 | Mask3D | 2 | 1 | 48.0(+2.0%) | 36.2(+2.0%) |
| **3DGraphLLM-2** | **Mask3D** (+ NMS) | **2** | **1** | **48.1**(+2.1%) | **36.5**(+2.3%) |
| 3DGraphLLM-0 | OneFormer3D | 0 | - | 45.4 | 34.5 |
| 3DGraphLLM-2 | OneFormer3D | 2 | 0 | 47.1(+1.7%) | 35.7(+1.2%) |
| **3DGraphLLM-2** | **OneFormer3D** (+NMS) | **2** | **1** | **47.5**(+2.1%) | **36.1**(+1.6%) |

version of 3DGraphLLM (3DGraphLLM-0) where the scene is represented as a sequence of object identifiers and features extracted by the 2D Object Encoder and the 3D Object Encoder, following the same training pipeline as 3DGraphLLM (3DGraphLLM-2) with two nearest neighbors. The 3DGraphLLM version with zero nearest neighbors serves as a baseline, equivalent to the Chat-Scene approach, which uses the same LLM as 3DGraphLLM. As shown in Table 3, incorporating a scene graph representation significantly improves the performance of the LLMs across all three 3D Vision-Language tasks: visual grounding, scene description, and question answering. However, the effect is more noticeable for the more modern LLAMA3-8B-Instruct. The pre-training on GT instance segmentation data improves the quality of the 3D Referred Object Grounding for LLAMA3-8B-Instruct and Vicuna-1.5-7B. For LLM Vicuna-1.5-7B, pre-training increases the Scene Captioning quality. For LLAMA3-8B-Instruct, pre-training improves the question answering on the Sqa3D dataset. The most interpretable metrics for the role of semantic edges are the accuracy metrics in the 3D Referred Object Grounding problem, so we keep this pre-training as part of the 3DGraphLLM training pipeline.

## 4.3 ABLATION STUDIES. 3D SCENE GRAPH REPRESENTATION

We conduct a series of experiments to explore methods for constructing a scene graph representation from a point cloud. In these experiments, we use a frozen version of LLAMA3-8B-Instruct (AI@Meta, 2024), training only the projection layers. We do not introduce new object tokens into the LLM's dictionary and follow a three-stage training process, including 3D Object Alignment, 3D Scene Alignment, and Instruction Tuning, as outlined in Chat3D (Wang et al., 2023a).

**Quality of instance segmentation.** We evaluate how the quality of scene segmentation into objects impacts the performance of 3DGraphLLM. As shown in Table 4, even with noisy neural network segmentation, representing the scene as a graph with semantic relationships is still more effective than using a simple list of objects. We conduct experiments with different object proposal methods, including OneFormer3D (Kolodiazhnyi et al., 2024) and Mask3D (Schult et al., 2023), but found no significant difference between them for our task. Therefore, in subsequent experiments, we use the Mask3D method to maintain consistency with the baseline Chat3Dv2 approach.

Neural network segmentation imperfections impact both the quality of object embeddings generated by the 3D Object Encoder and the embeddings of semantic relations between objects. We perform a PCA analysis of Uni3D object embeddings and VL-SAT relation embeddings, comparing results for ScanNet training scenes using GT instance segmentation and Mask3D instance segmentation (see Appendix A). Our analysis shows that, with the standard selection of nearest neighbors, the relation embeddings differ significantly between GT and Mask3D three-dimensional masks.

By examining the minimum distance between neighboring objects, we observed that duplicate objects were often selected as neighbors. To address this issue, we introduced a minimum distance filter of 1 cm between neighboring objects, which made the relationship embeddings from GT masks and Mask3D results more consistent. Additionally, applying this filter improved performance on the visual grounding task, as shown in Table 4. We also experimented with adding an NMS filter to remove duplicates among the vertices that an object may be associated with, with a threshold of $IoU = 0.99$. The results in Table 4 show that adding the filter allows for further improvement of the grounding quality.

**Number of nearest neighbors.** We conducted an experiment to examine how the number of nearest neighbors affects the quality of visual grounding and the speed of model inference, as adding more connections increases the number of tokens used to describe each object. This experiment was performed using ground-truth scene segmentation and the RioRefer dataset (Miyanishi et al., 2024), as this setup provides the highest quality embeddings for semantic relations between objects. We vary the number of nearest neighbors in powers of two, capping it at 5 due to GPU memory constraints during training. As shown in Figure 4, increasing the number of nearest neighbors enhances visual grounding quality with a slight increase in inference time.

**Spatial relations.** Previous research (Wang et al., 2023a; Huang et al., 2023) has shown that incorporating spatial relationships between objects, represented by 3D coordinates of their bounding boxes, can improve performance in visual grounding tasks. We attempted to integrate spatial relations into our method by using the output of the spatial transformer as the final token in the relation triplets between an object

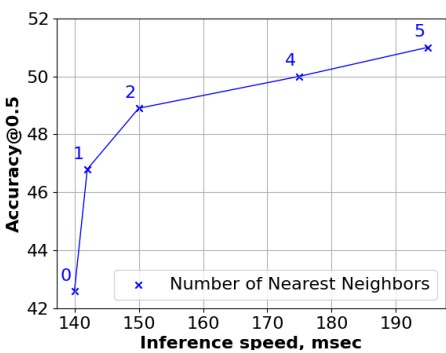

Figure 4: Dependence of inference speed and visual grounding accuracy on the number of nearest neighbors in the object subgraph. This experiment utilizes the RioRefer dataset along with GT instance segmentation.

Table 5: Ablation study on spatial relation module on RioRefer dataset (GT Instance segmentation).

| Methods | Edge Number | Spatial relation | Acc@0.5↑ |
|---|---|---|---|
| 3DGraphLLM | 0 | ✓ | 42.6 |
| 3DGraphLLM | 2 | ✓ | 48.9(+6.3%) |
| 3DGraphLLM | 2 | ✗ | **50.1**(+7.5%) |

and its nearest neighbors (i.e., a triplet $(F_i^v, F_{ij_k}^e, F_{j_k}^{\text{rel}})$, where $F_{j_k}^{\text{rel}}$ represents the output of the Chat3Dv2 spatial relation module (Huang et al., 2023)). However, as shown in Table 5, our experiments did not find this approach effective for learning a graph representation of a scene.

## 5 CONCLUSION

In this paper, we propose a new learnable approach to using a 3D semantic scene graph for a large language model solving the 3D vision-language tasks. Detailed experiments demonstrate the effectiveness of this approach, which explicitly takes into account semantic relations between objects represented as 3D point clouds. Our approach, called 3DGraphLLM, demonstrated state-of-the-art quality on popular ScanRefer, Multi3DRefer, and Scan2Cap datasets.

A limitation of the method is a significant increase in resource consumption with an increase in the edge number for each graph node. At the same time, we showed that taking into account only two edges for each object demonstrates an acceptable trade-off between performance and model quality.

For further development of the work, it seems appropriate to search for the methods to reduce token usage for encoding object relationships in our graph representation. Another important aspect for further work is the creation of methods for generating semantic relations between objects that are robust to imperfections in the instance segmentation of the scene point cloud.

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

## A PCA ANALYSIS OF UNI3D OBJECT EMBEDDINGS AND VL-SAT RELATION EMBEDDINGS

We conduct a PCA analysis of Uni3D object embeddings and VL-SAT relation embeddings, comparing results on ScanNet training scenes using both GT instance segmentation and Mask3D instance segmentation.

Our findings indicate that the relation embeddings exhibit notable differences between GT and Mask3D three-dimensional masks when the naive nearest-neighbor selection is applied. When applying the minimal distance filter, the similarity of VL-SAT relation embeddings significantly increases between GT and Mask3D instance segmentation.

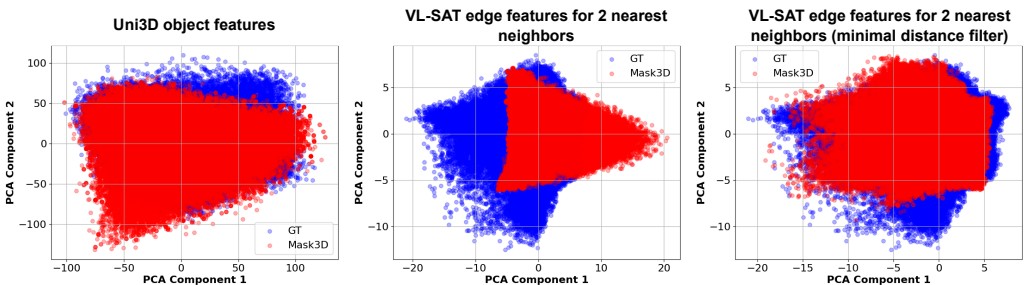

Figure 5: Comparison of Uni3D object features and VL-SAT semantic edge features for the two nearest neighbors (NNs) based on ground-truth (GT) scene segmentation and Mask3D scene segmentation within the ScanNet training set. **Left**: Uni3D object features are relatively close for GT point clouds and Mask3D point clouds. **Center**: using the standard approach for selecting NNs to generate VL-SAT features, the features for pairs of Mask3D point clouds differ significantly from those of GT point clouds. **Right**: after applying a minimum neighbor distance filter for selecting NNs, the VL-SAT features for object pairs from Mask3D instance segmentation align more closely with those from GT instance segmentation.

## B COMMON FAILURE CASES

We illustrate the most common failure cases of 3DGraphLLM related to spatial relationships in Figure 6.

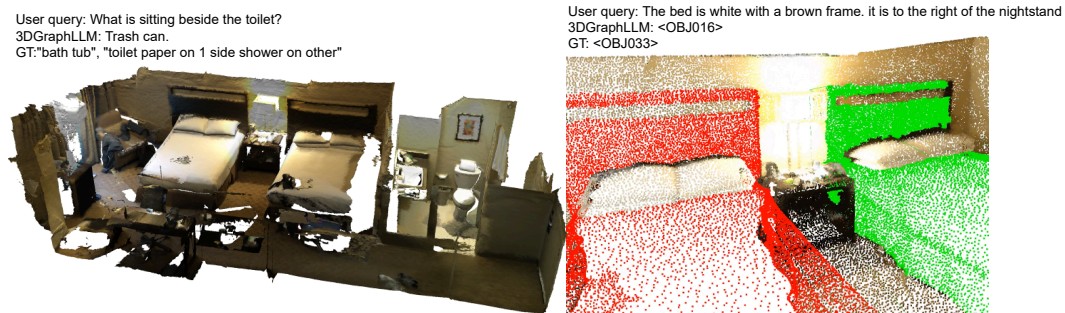

Figure 6: Common failure cases of 3DGraphLLM related to spatial relationships. **Left**: In the ScanQA dataset, 3DGraphLLM incorrectly identifies the front/back and left.right directions relative to the observer. **Right**: In the ScanRefer dataset, 3DGraphLLM confuses left and right. The GT object is highlighted in green, and the 3DGraphLLM prediction is highlighted in red.

## C  FUNCTIONAL QUERIES

We illustrate the ability of 3DGraphLLM to leverage common sense knowledge in its responses to question types not present in the training dataset in Figure 7.

User query: Name the type of room described by the list of objects.
3DGraphLLM: Hotel room.
User query: Can I make dinner in this room?
3DGraphLLM: No.

User query: What object can I use to do my homework?
Answer with object ID.
3DGraphLLM: <OBJ037>.

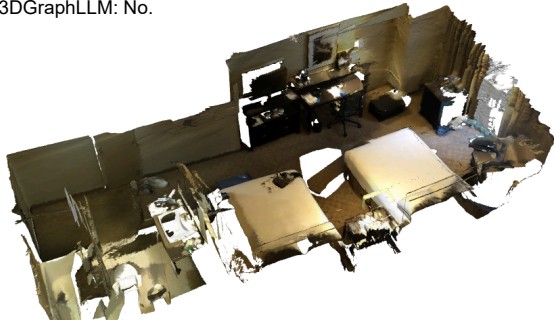
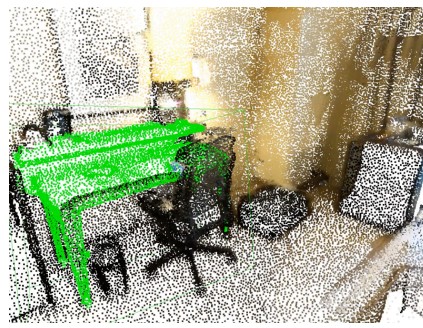

Figure 7: Functional queries about the room and objects to the 3DGraphLLM. **Left**: 3DGraphLLM is capable of answering questions about functional properties of the room and its room type. **Right**: 3DGraphLLM is capable of answering questions about the functional properties of objects in a room.

