# OpenReview forum: "3DGraphLLM: Combining Semantic Graphs and Large Language Models for 3D Referred Object Grounding"
_ICLR.cc/2025/Conference — Submitted to ICLR 2025_

### Official Review · Reviewer_cnDL · 2024-11-01

**Soundness:** 3
**Presentation:** 3
**Contribution:** 2
**Rating:** 6
**Confidence:** 4

**Summary:**

This paper proposes 3DGraphLLM, a multimodal language model that takes into account the object relationships in a 3D scene and achieves better performance in referred object grounding tasks. 3DGraphLLM takes as input a scene point cloud, segmented into object instances using an off-the-shelf 3D object segmentation model. The features for each 3D object and their relationships are also extracted by pretrained models, projected to LLM token space, and input to the LLM as part of the prompt. The LLM and the projectors are finetuned on the training sets of 3D datasets. Experiments show that 3DGraphLLM outperforms baselines and several ablation studies show the effectiveness of design decisions.

**Strengths:**

1. 3DGraphLLM converts object relationships to LLM token space and incorporates them into the prompt for 3D grounding tasks. In this way, the LLM receives more complete information about the 3D scene graph and gets a better understanding of the 3D environment.
2. Most design decisions have ablation studies and their effectiveness is verified.
3. The paper is well-written and easy to follow.

**Weaknesses:**

1. LLMs are known for capturing a significant amount of common sense knowledge and show out-of-distribution generalization ability. However, in the design of 3D GraphLLM, I cannot see how such power of LLMs is being utilized or demonstrated at all, and thus **what's the point of using a pretrained LLM there?** Ideally, by using an LLM, I expect to see that in addition to taking the new modality (3D scene graph) as input, the adapted LLM also shows that its common sense knowledge is also being exploited in a non-trivial way, e.g. out-of-distribution generalization, achieving tasks other than grounding. At least I want to see if the entire model is trained from scratch (without pretraining on text), and whether the performance will be worse.

With the above being said, I'm very concerned that in Table 2, the proposed method has worse performance than the expert models on ScanRefer, which seems to suggest that the LLM pretraining is not really helping.

2. The proposed method relies heavily on the performance of the underlying 3D segmentation and scene graph generation models used to encode 3D objects and relationships into features. Even if the LLM is able to generalize OOD, it's still limited to these models and cannot recover from their failures.

This concern is verified by Table 4, where using GT segmentation, 3DGraphLLM is better than the baseline by 5.6 percent, but less than 2.0 percent using predicted instance segmentation.

3. Why is the flat graph representation object-centric (i.e. obj1, obj1-NN1, obj1-NN2, obj2, obj2-NN1, obj2-NN2...), instead of further flattened (i.e. obj1, obj2, obj3, ..., rel1, rel2, rel3, ...)? In the latter format, the set of relationships to keep can still be limited to the k-NN of each object instead of the dense graph. I'm asking because in Table 1 (L273), seems the $F_1^v$ is being repeated k times. The repetition will be even greater if one object is considered as NN of other object nodes. Therefore, there seems to be still a lot of waste in the proposed tokenization scheme.

4. This paper proposes to train the model in a two-stage manner (first on GT instance segmentation, then on predictions), but this is not being ablated.

5. The design minimum distance filter suggests that there are duplicate detections in the 3D detection results. I think these duplicates should be handled using filtering techniques like 3D NMS rather than the proposed naive minimum distance filter on the scene graph edges.

**Questions:**

Please see the "Weakness" section.

---

> ### Author Response · Authors · 2024-11-23
>
> Thank you for your valuable feedback. We have addressed the raised concerns by modifying the sequence generation method describing an object, adding a 2D object token alongside the object identifier token. This modification, inspired by the Chat-Scene method [1], improved the stability of the 3DGraphLLM's convergence without altering the core contribution of this work: developing a method for creating a trainable representation of the 3D semantic scene graph and an algorithm for generating a flat sequence from the graph based on the object's k-nearest neighbors. Additionally, we conducted further experiments with the NMS filter and found that it provides an additional improvement in the quality of duplicate object filtering.
>
> > LLMs are known for capturing a significant amount of common sense knowledge and show OOD generalization ability. ... Ideally, by using an LLM, I expect to see that in addition to taking the new modality (3D scene graph) as input, the adapted LLM also shows that its common sense knowledge is also being exploited in a non-trivial way, e.g. OOD generalization, achieving tasks other than grounding. ...
>
> The revised version of 3DGraphLLM shows a significant boost in the performance of 3D Referred Object Grounding, Scene Captioning, and Question Answering (QA) tasks compared to baseline expert models that do not use pre-trained LLMs, including those that rely on multi-task learning (3D-Vista, PQ3D). As shown in Table 2 in the general response, methods based on LLMs significantly outperform expert models in the Situated Question Answering task (SQA3D). Finally, the use of LLMs enables answering various types of queries, including those requiring common sense knowledge (see Appendix Section C in the revised version of the paper).
>
> > The proposed method relies heavily on the performance of the underlying 3D segmentation and scene graph generation models used to encode 3D objects and relationships into features...
>
> Indeed, the results of our experiments show that the greatest impact of using a 3D semantic scene graph is observed in when using GT instance segmentation of 3RScan scene dataset (up to an 8% improvement in object grounding). However, using pre-trained models for object proposals is a common approach for solving 3D Vision-Language tasks. For example, this approach is used in methods like Chat-3D-v2, Grounded 3D-LLM, and Chat-Scene. Despite the dependence on the quality of 3D segmentation, such methods, including 3DGraphLLM, show significantly better results than methods using voxel (Scene-LLM) or point cloud (3D-LLM) representations of the scene. Combined with nearest neighbor filtering and LoRa fine-tuning, 3DGraphLLM shows an improvement of about 3% for the 3D object grounding task, as well as an increase in the quality of Scene Captioning (+3-4% CIDEr@0.5), while maintaining comparable quality in the QA task (see Table 1 in the general response).
>
> > Why is the flat graph representation object-centric ..., instead of further flattened...? ... Therefore, there seems to be still a lot of waste in the proposed tokenization scheme.
>
> When creating an object-centric approach to the flat representation of a scene graph, our motivation was based on two points:
> - The object-centric scene representations work well when used as input to LLM(Grounded 3D-LLM, Chat3Dv2, Chat-Scene).
> - We wanted to provide the LLM with not only the type of relationship between two objects but also information about which two objects the relationship connects.
>
> In the naive approach, where we simply take the semantic relationship tokens and move them to the end of the object list, we lose information about which specific two objects are connected by the relation rel1. However, your comment is valid, and one direction of future work may be to reduce the number of tokens needed to encode the relationships between objects in our graph representation. We have added this direction for future work to the conclusion of the revised version of the article.
>
> > This paper proposes to train the model in a two-stage manner ..., but this is not being ablated.
>
> We added an ablation for training in a two-stage manner, which shows that using pre-training with GT instance segmentation improves the performance of the 3D Referred Object Grounding task, while keeping the solution for the Scene Captioning task better than the baseline method, where no edges between objects are present. Please refer to Table 1 in the general response.
>
> > ... I think these duplicates should be handled using filtering techniques like 3D NMS rather than the proposed naive minimum distance filter on the scene graph edges.
>
> Thank you for the comment. We added the NMS filter to our pipeline, which resulted in an additional performance boost (see Table 3 in the general response).
>
> [1] Huang, H., et al. (2024). Chat-scene: Bridging 3d scene and large language models with object identifiers. In The 38 Annual Conference on Neural Information Processing Systems.

---

> > ### Comment · Reviewer_cnDL · 2024-11-25
> >
> > Thanks for the response!
> >
> > I wonder for "in the naive approach, where we simply take the semantic relationship tokens and move them to the end of the object list, we lose information about which specific two objects are connected by the relation rel1", have you tested this variant and got ablation results?
> >
> > I'm asking because changing the order of the object and relation tokens in an LLM input may not harm the results, as long as they are consistent in training and testing. Positions in LLMs are purely denoted by the positional encoding. Non-adjacent tokens in the input can also be attended in a transformer (it may require a proper separator token). The current design wastes the input tokens, while not showing necessity.

---

> > ### Comment · Reviewer_cnDL · 2024-11-25
> >
> > For the OOD generalization comment, I'm still interested in seeing the result on "if the entire model is trained from scratch (without pretraining on text), and whether the performance will be worse". Because I'm still not clear at the moment how much advantage is brought by LLM and pretraining to text.
> >
> > Also please clearly note what "pre-train" means in Table 1 inthe  general response.

---

> > > ### Author Response · Authors · 2024-11-27
> > >
> > > Please note that training an 8-billion-parameter model from scratch requires a significant amount of computational resources, which we do not have. Previous works, such as 3D-vista [1], have conducted experiments on scaling transformer models for 3D Vision-Language tasks. Their results indicate that increasing the depth of the transformer model performing multimodal fusion does not improve the performance of 3D Vision-Language tasks. The authors attribute this to the fact that existing datasets for 3D scenes are still substantially limited in size, preventing the scaling effects observed in NLP and 2D tasks. Therefore, we believe that 3DGraphLLM fully leverages the properties of the pre-trained LLM, and we assume that training a model from scratch using the currently available 3D scene datasets would result in noticeably lower performance and шt would have required significantly more computational resources and made it more difficult for other researchers to reproduce our results.
> > >
> > > [1] Zhu, Z., Ma, X., Chen, Y., Deng, Z., Huang, S., & Li, Q. (2023). 3d-vista: Pre-trained transformer for 3d vision and text alignment. In Proceedings of the IEEE/CVF International Conference on Computer Vision (pp. 2911-2921).

---

> > > > ### Comment · Reviewer_cnDL · 2024-11-28
> > > >
> > > > Thanks authors for the efforts in the responses!
> > > >
> > > > My concerns are mostly addressed and I have raised my rating to 6.
> > > >
> > > > One more possible concern is that the addition of 2D tokens is a major thing that happens after the initial submission, causes a huge difference from the initial results, and is not something brought/requested by the reviewers. I will leave this to AC to decide whether this should be an issue.

---

> > > > > ### Author Response · Authors · 2024-11-30
> > > > >
> > > > > Thank you for the feedback.

---

> ### Author Response · Authors · 2024-11-27
>
> Thank you for your response! Yes, we conducted this experiment during the rebuttal period. We removed repeated object tokens but kept the relation tokens next to the object tokens to ensure the approach also works in cases where the number of objects differs between scenes. That is, the sequence is the following:
>
> $[<OBJ001> F_{1}^{2d}, F_{1}^{v}, F_{12}^{e}, F_{14}^{e}, ...<OBJN>F_{N}^{2d}, F_{N}^{v}, F_{Nk_1}^{e}, F_{Nk_2}^{e}$]
>
>
> Below is a table comparing this approach (second row) with the approach described in the paper, where relations are represented as triplets (third row). As seen from the table, using relations in the form of triplets improves the quality of 3D Referred Object Grounding task performance.
>
> | Methods | Pre-train | Relations as triplets | Number of edges | ScanRefer A@0.5↑ | Multi3DRefer F1@0.5 | ScanQA C↑ | ScanQA B-4↑ | Sqa3D EM↑  | Scan2Cap C@0.5↑ | Scan2Cap B-4@0.5↑ |
> |-------------|------------|------------|----------------------|------------------------|--------------------------|---------------|----------------|---------------|---------------------|-----------------------|
> | 3DGraphLLM-0 LLAMA3-8B | ✗ | ✗ | 0 | 52.0 | 55.1 | 84.0 | 15.8 | 53.8 | 80.0 | 37.5 |
> | 3DGraphLLM-2 LLAMA3-8B | ✗ | ✗ | 2 | 54.2 | 56.3 | 85.6 | 15.1 | 54.6 | 87.2 | 39.3 |
> | 3DGraphLLM-2 LLAMA3-8B | ✗ | ✓ | 2 | 54.3 | 57.3 | 87.4 | 14.9 | 54.5 | 85.6 | 39.6 |

---

> > ### Comment · Reviewer_cnDL · 2024-11-28
> >
> > Thanks a lot for providing these results!
> >
> > I think the results are mixed - adding relations as triplets is not consistently better but causes a waste. I suggest authors list this result in the paper for completeness.

---

> > > ### Author Response · Authors · 2024-11-30
> > >
> > > Thank you for the response! We will include this result in the final version of the paper.

---

### Official Review · Reviewer_aDrA · 2024-11-01

**Soundness:** 3
**Presentation:** 3
**Contribution:** 2
**Rating:** 6
**Confidence:** 4

**Summary:**

The paper considers the problem of generating a learnable representation of a 3D scene to be processed by Large Language Models (LLMs). Compared to existing object-centric latent scene representations, the proposed approach incorporates an additional relational edge encoding. Both node (objects) and edge (relations) encodings are obtained by passing the off-the-shelf point cloud encoder outputs through a trainable projection head. Experiments show that this modification improves the performance on visual grounding tasks across multiple datasets.

**Strengths:**

1. The idea to add object semantic relationship into the latent scene representation for LLMs is well-motivated.
2. The proposed approach is simple and effective.

**Weaknesses:**

The proposed modification introduces a parameter of minimum distance for filtering the nearest neighbors. It requires manual tuning, as is set to be 1 cm as in the paper. This requirement might potentially make it hard to apply this approach to new domains, such as a tabletop scene or a larger multi-floor multi-room indoor scene.

**Questions:**

1. What is the difference between the “3DGraphLLM-0” (in table 3, 4) and “Chat-3D v2” baseline (in table 2)? It is said in Section 4.4 that they are equivalent (“The 3DGraphLLM version with zero nearest neighbors…, equivalent to the Chat3Dv2 approach..”). However, under the ScanRefer benchmark, the 3DGraphLLM-0 (in table 4) outperforms Chat3Dv2 (in table 2) by more than 10% margin in $Acc\@0.25$. Similarly, why is the performance of 3DGraphLLM-0 under GT segmentation not consistent in table 3 and 4?
2. The proposed approach uses Uni3D for object encoding and VL-SAT for edge encoding. Since VL-SAT also contains an object encoder, which might be even more compatible with its own edge encoder,  why is the Uni3D encoder used instead?
3. What are the common failure cases of the method?

---

> ### Author Response · Authors · 2024-11-23
>
> Thank you for your valuable feedback. We would like to address the raised concerns:
>
> > The proposed modification introduces a parameter of minimum distance for filtering the nearest neighbors. It requires manual tuning, as is set to be 1 cm as in the paper. This requirement might potentially make it hard to apply this approach to new domains, such as a tabletop scene or a larger multi-floor multi-room indoor scene.
>
> The primary contribution of our work is a method for creating a learnable representation of the scene graph for LLMs, which improves the performance of 3D Vision-Language tasks compared to methods that do not leverage semantic relationships between objects. When using ground-truth point cloud segmentation, 3DGraphLLM does not require a distance-based nearest neighbor filter. The motivation for selecting a distance threshold is solely to filter out duplicate objects. The minimum distance between two ground-truth objects in the ScanNet dataset is approximately 5 cm, so we expect that a 1 cm distance filter is sufficiently lenient.
> We also conducted experiments with the Non-Maximum Suppression (NMS) filter, which improved the proposed duplicate object filtering method (see Table 3 in the the general response) and can be independently applied in other domains. Furthermore, if it is possible to construct a higher-quality scene graph in a new domain, additional nearest neighbor filtering might not be necessary.
>
> > What is the difference between the “3DGraphLLM-0” (in table 3, 4) and “Chat-3D v2” baseline (in table 2)? It is said in Section 4.4 that they are equivalent (“The 3DGraphLLM version with zero nearest neighbors…, equivalent to the Chat3Dv2 approach..”). However, under the ScanRefer benchmark, the 3DGraphLLM-0 (in table 4) outperforms Chat3Dv2 (in table 2) by more than 10% margin in . Similarly, why is the performance of 3DGraphLLM-0 under GT segmentation not consistent in table 3 and 4?
>
> The Chat3Dv2 baseline in Table 2 uses PointGroup as the instance segmentation method, as well as the Vicuna-7B LLM, and a slightly different training procedure. 3DGraphLLM-0 corresponds to the Chat3Dv2 version with the same training parameters as 3DGraphLLM: Mask3D as the instance segmentation method, LLAMA3-8B-Instruct LLM, and a two-stage training procedure. Table 4 differs from Table 3 in that it pertains to the Ablation study series in Section 4.4.1, where we use the frozen version of the LLM without LoRa fine-tuning, and do not introduce object identifier tokens.
> Note that in the revised version of the paper, Table 3 corresponds to the ablation study for different LLM models and GT pretraining. In the revised version of the paper, the closest baseline to 3DGraphLLM is Chat-Scene[1].
>
> > The proposed approach uses Uni3D for object encoding and VL-SAT for edge encoding. Since VL-SAT also contains an object encoder, which might be even more compatible with its own edge encoder, why is the Uni3D encoder used instead?
>
> VL-SAT was developed to solve the task of generating a 3D semantic scene graph in the domain of the 3RScan scenes. It uses a fixed set of classes for vertex and edge classification along with the ground truth instance segmentation of the scene. Uni3D was trained on a large corpus of 1M point clouds, aligning point cloud representations with text descriptions and images, which is why we assume it has better generalization to new domains compared to VL-SAT object encoder. We expect that the compatibility of the vertices and edges representations is achieved through the training of projection layers and model LoRa fine-tuning. However, we consider as the future work the research on scene graph construction methods that do not rely on ground truth segmentation of the scene. These methods may produce consistent representations of point clouds and the relationships between them.
>
> > What are the common failure cases of the method?
>
> A common failure case for our method occurs when answering questions that involve semantic relationships between objects that depend on the viewpoint (e.g., left/right, front/behind). Please refer to Appendix Section B in the revised version of the paper.
>
> [1] Huang, H., Chen, Y., Wang, Z., Huang, R., Xu, R., Wang, T., ... & Zhao, Z. (2024). Chat-scene: Bridging 3d scene and large language models with object identifiers. In The Thirty-eighth Annual Conference on Neural Information Processing Systems.

---

> > ### Comment · Reviewer_aDrA · 2024-11-30
> >
> > Thanks to the authors for the efforts. The explanations and the additional experiments address most of my questions. I would like to raise my score to 6.

---

> > > ### Author Response · Authors · 2024-11-30
> > >
> > > Thank you for the feedback!

---

### Official Review · Reviewer_9KjD · 2024-11-02

**Soundness:** 3
**Presentation:** 3
**Contribution:** 3
**Rating:** 6
**Confidence:** 5

**Summary:**

The paper introduces a novel method called 3DGraphLLM, which combines semantic graphs and large language models (LLMs) for 3D referred object grounding in scenes. The method constructs a learnable representation of a 3D scene graph to be used as input for LLMs to perform 3D referred object grounding tasks. Experiments demonstrate its advantages over baseline methods that do not utilize semantic relationships between objects on popular datasets like ScanRefer, RIORefer, and Multi3DRefer.

**Strengths:**

1. This paper notes that the semantic relationship between object pairs within a 3D scene is beneficial for visual grounding and other tasks such as visual captioning and visual question answering. It proposes 3DGraphLLM, which constructs a learnable representation of a 3D scene graph and uses it as input to a large language model (LLM) to perform the 3D referred object grounding task.

2. The authors have identified some technical issues (such as the domain gap between 3Rscan and ScanNet, and the presence of duplicate objects in the nearest neighbors), and I believe this paper is technically thorough.

3. The authors provide relatively sufficient and convincing experimental results.

**Weaknesses:**

1. In the Edge Feature Encoder, the important issue is raised that the semantic relationships recognition methods are all trained on 3RScan scenes, while the visual grounding tasks are typically tested on ScanNet scenes. I believe this is indeed a challenge. The authors claim that the solution is to use VL-SAT due to its good generalization in the new domain. I am very familiar with VL-SAT, and currently, there is no evidence showing that this method has good generalization in the new domain to address the mentioned challenge.

2. Similarly regarding VL-SAT, the authors seem to have used the VL-SAT pre-trained on 3RScan to test on the ScanNet dataset (e.g., ScanRefer, Multi3DRefer, etc.). This is equivalent to the results of RIORefer->ScanRefer in Cross3DVG. In Figure 5, when the number of nearest neighbors = 2, the Acc@0.5 reaches an astonishing 49.0, which not only significantly surpasses the result of RIORefer->ScanRefer in Cross3DVG (13.34) but also shows a remarkable difference compared to the RIORefer->RIORefer test result (20.21) given in Cross3DVG. This could be explained by 1) using GT instance segmentation yields better results, and 2) 3DGraphLLM performs exceptionally well on RIORefer, surpassing the Expert model. However, 1) based on the results in Table 4, it is difficult to foresee such a significant gap, and 2) the performance of the model on similar datasets like ScanRefer and Multi3DRefer indicates that it is challenging to significantly exceed the existing SOTA Expert model. I hope the authors can provide some explanations regarding the above points and present more experimental results on RIORefer during the rebuttal period. A reasonable response to this weakness would make me consider raising my score.

3. Table 4 needs to be completed (i.e., providing the results of 3DGraphLLM-2 with OneFormer3D segmentation when the minimal distance = 1cm), as this is crucial for validating the effectiveness of the minimal distance strategy.

4. The results of VL-SAT are based on training with 160 objects and 27 relationships. I would like to know how much these 27 relationships help in ScanRefer. As far as I know, these 27 relationships do not seem to have a strong correlation with the descriptions in ScanRefer. I understand that this is essentially due to the limitations of dataset annotations, but I hope the authors can explore whether the annotations in the 3D Scene Graph Generation dataset are related to the descriptions in visual grounding.

**Questions:**

1. Please address the questions in Weaknesses 1. I have concerns about the generalization between the two datasets.

2. Please respond to the questions in Weaknesses 2. I would like to see more experimental results on RIORefer and hope the authors can explain the significant differences in the results I mentioned in Weaknesses 2.

3. I hope the authors can answer the questions in Weaknesses 4, as I believe the correlation between the two datasets is very important.

---

> ### Author Response · Authors · 2024-11-25
>
> Thank you for your feedback. We would like to address your comments:
>
> > In the Edge Feature Encoder, the important issue is raised that the semantic relationships recognition methods are all trained on 3RScan scenes, while the visual grounding tasks are typically tested on ScanNet scenes. ... The authors claim that the solution is to use VL-SAT due to its good generalization in the new domain. I am very familiar with VL-SAT, and currently, there is no evidence showing that this method has good generalization in the new domain to address the mentioned challenge.
>
> In our experiments, when using GT instance segmentation, incorporating VL-SAT edge features in our 3DGraphLLM method improves performance in the 3D Referred Object Grounding task by +5.6% on the ScanNet scene dataset (ScanRefer dataset, Table 4) and +7.5% on the 3RScan scene dataset (RioRefer dataset, Table 5). Although the use of VL-SAT semantic edges is slightly less effective for ScanNet scenes compared to 3RScan scenes, this experiment demonstrates that VL-SAT exhibits good generalization across the two scene datasets for our task. This may be due to the fact that we use VL-SAT edge features before their classification into 27 categories.
>
> > Similarly regarding VL-SAT, the authors seem to have used the VL-SAT pre-trained on 3RScan to test on the ScanNet dataset. This is equivalent to the results of RIORefer->ScanRefer in Cross3DVG. In Figure 5, when the number of nearest neighbors = 2, the Acc@0.5 reaches an astonishing 49.0, which not only significantly surpasses the result of RIORefer->ScanRefer in Cross3DVG (13.34) but also shows a remarkable difference compared to the RIORefer->RIORefer test result (20.21) given in Cross3DVG. .... I hope the authors can provide some explanations regarding the above points and present more experimental results on RIORefer during the rebuttal period....
>
> As we understand Table 2 in the Cross3DVG paper, the values Acc@0.5 13.34 and 20.21 correspond to the ScanRefer->RIORefer and ScanRefer->ScanRefer settings, respectively. Given this, 3DGraphLLM indeed outperforms the expert model from Cross3DVG in the ScanRefer->ScanRefer setting. Moreover, in the revised version of the paper, 3DGraphLLM also surpasses other expert models in the ScanRefer->ScanRefer setting (see Table 2 in the general response). Unfortunately, conducting experiments with instance segmentation on the 3RScan scene dataset requires additional work to verify the generalization capability of Mask3D to 3RScan. Therefore, we leave these experiments for future work.
>
> > Table 4 needs to be completed (i.e., providing the results of 3DGraphLLM-2 with OneFormer3D segmentation when the minimal distance = 1cm), as this is crucial for validating the effectiveness of the minimal distance strategy.
>
> Thank you for your comment. We conducted this experiment in the updated version of the paper. This experiment demonstrates that the combination of the NMS filter and the minimum distance filter is also effective for OneFormer3D:
>
> | Methods | Instance segmentation | Number of edges | Minimal distance (cm) | ScanRefer A@0.25↑ | ScanRefer A@0.5↑ |
> | ---------------- | ------------------------- | ------------------- | ------------------------- | ----------------------- | ---------------------- |
> | 3DGraphLLM-0 | OneFormer3D | 0 | - | 45.4 | 34.5 |
> | 3DGraphLLM-2 | OneFormer3D  | 2 | 0 | 47.1 (+ 1.7%) | 35.7 (+ 1.2%) |
> | 3DGraphLLM-2 | OneFormer3D  (+ NMS) | 2 | 1 | 47.5 (+ 2.1%) | 36.1 (+1.6%) |
>
> > The results of VL-SAT are based on training with 160 objects and 27 relationships. I would like to know how much these 27 relationships help in ScanRefer. ... I hope the authors can explore whether the annotations in the 3D Scene Graph Generation dataset are related to the descriptions in visual grounding.
>
> We calculated the ratio of annotations in ScanRefer where object descriptions do not include any of the 27 VL-SAT relations. We use two methods to identify these descriptions. “Exact matching” refers to the precise detection of mentions of one of the 27 relations in the object descriptions in ScanRefer. As shown in the table below, in this case, more than half of the annotations in ScanRefer contain at least one of the 27 VL-SAT relations. Since we use a pre-trained LLM and VL-SAT hidden edge features, we also investigate how many ScanRefer annotations include synonyms of the 27 VL-SAT relations. We consider closely related synonyms such as "under," "on," "at," "is mounted," "is affixed," "near," and "next." Accounting for these spatial relations, the proportion of descriptions containing spatial relations different from 27 VL-SAT relations reduces to 12% in both the training and validation sets.
>
> | Split | Searching method | Ration of descriptions without VL-SAT relations, % |
> | ---- | ----------------------- | ---------------- |
> | train | with synonyms | 12.2% |
> | train | exact matching | 43.4% |
> | val | with synonyms | 12.3% |
> | val | exact matching | 41.6% |

---

> ### Comment · Reviewer_9KjD · 2024-11-27
>
> Thank you to the authors for their response. My concerns have been addressed, and I would like to raise my score to 7. However, since there is no option for that and only an 8 is available, I feel that an 8 is not appropriate for this paper. Therefore, I will maintain my current score and increase my confidence level to 5. Wishing you the best of luck!

---

> > ### Author Response · Authors · 2024-11-27
> >
> > Thank you for your feedback!

---

### Official Review · Reviewer_qz74 · 2024-11-04

**Soundness:** 2
**Presentation:** 3
**Contribution:** 2
**Rating:** 3
**Confidence:** 4

**Summary:**

3DGraphLLMs proposes to improve referential grounding performance of LLM based methods by supplying an explicit spatial graph representation of the scene as an input to the LLM (in addition to the task and the object information). Specifically, the paper proposes to model relationships between objects using an off-the-shelf 3D scene graph model. Crucially, these relationships are only modelled among k nearest neighbors of the object to reduce the number of input tokens to the LLM. To prevent very neaby object proposals to be considered as neighbors, the paper use heurestics such as ignoring boxes that are too close to each other (1cm threshhold). The proposed model is tested on ScanRefer and Multi3DRefer grounding datasets and obtain better performance than other LLM based grounding models.

**Strengths:**

The paper is well-written and easy to follow. It adds onto an interesting line of work in using LLMs for 3D referential grounding, The ablation study is well-designed and covers several of the relevant questions about design choices.

**Weaknesses:**

The quantitative numbers of the papers are weak:
- The LLM used by this paper is the recent LLAMA3-8B-Instruct while other baselines use different and older LLMs. for eg. Chat3Dv2 and Grounded 3D LLM uses Vicuana-7B (based on LLAMA 2). This makes the comparison unfair with prior models.
- The paper does not report results on other benchmarks like ScanQA, SQA3D and Scan2Cap while still training on it, and showing ablations on it. The paper mentions that the focus is on visual grounding tasks, and that is why the evaluations are only shown in these benchmarks; however the key idea of using scene graph representation might also help for these other spatial reasoning tasks and if not, should atleast be reported for completeness in my opinion.
- The boost in performance by using the proposed scene graph technique is only around 1-2% (Table-4) in the setup when models uses masks from a detector like Mask3D (which represents the realistic setup). This boost comes at a cost of additional complexities of making the scene graphs and querying the LLMs with additional tokens (additional ~800 tokens per query).
- The paper claims to achieve SOTA results on ScanRefer and Multi3DRefer benchmarks in the introduction. These claims do not look correct -- as per Table-2, 3D Vista and M3DRef-CLIP outperforms the proposed model on ScanRefer. Besides several stronger baselines like BUTD-DETR [1] and PQ3D [2] outperform 3DGraphLLMs by large margins. Perhaps, the intention was to say that 3D GraphLLMs is SOTA LLM-based visual grounding model -- the claim in introduction may likely need to be revised appropriately. 3DGraphLLMs outperformM3DRef-CLIP baseline on Multi3DRefer benchmark. However, the baseline is trained on much lesser data compared to 3DGraphLLMs.

**Questions:**

The improvement from using 3DGraph representations seems somewhat marginal when compared in apples-to-apples setting with the predicted masks (Table-4). The results in Table-2 look good in the first glance, but due to the use of a stronger base LLM by the proposed method,  the results are hard to interpret directly. I would be happy to increase my rating if provided with more evidences / reasons of 3DGraphLLMs helping the grounding tasks (while not hurting the QA and captioning tasks by also reporting numbers on those).

---

> ### Author Response · Authors · 2024-11-23
>
> Thank you for your valuable feedback. We have addressed the raised concerns by modifying the sequence generation method describing an object, adding a 2D object token alongside the object identifier token. This modification, inspired by the Chat-Scene method [1], improved the stability of the 3DGraphLLM's convergence without altering the core contribution of this work: developing a method for creating a trainable representation of the 3D semantic scene graph and an algorithm for generating a flat sequence from the graph based on the object's k-nearest neighbors. Additionally, we conducted further experiments with the Non-Maximum Suppression filter and found that it provides an additional improvement in the quality of duplicate object filtering.
>
> >The LLM used by this paper is the recent LLAMA3-8B-Instruct while other baselines use different and older LLMs. for eg. Chat3Dv2 and Grounded 3D LLM uses Vicuna-7B (based on LLAMA 2). This makes the comparison unfair with prior models.
>
> We conducted an additional experiment with Vicuna1.5-7B (used by Chat-Scene[1] - the closest baseline for our method). As shown in Table 1 in the general response, 3DGraphLLM-Vicuna1.5-7B still provides an advantage compared to baseline methods. However, it is worth noting that semantic edges offer a greater quality boost for LLAMA3-8B-Instruct, which is why we use it in our method.
>
> >The paper does not report results on other benchmarks like ScanQA, SQA3D and Scan2Cap while still training on it, and showing ablations on it. The paper mentions that the focus is on visual grounding tasks, and that is why the evaluations are only shown in these benchmarks; however the key idea of using scene graph representation might also help for these other spatial reasoning tasks and if not, should atleast be reported for completeness in my opinion.
>
> We have updated Table 2 in the revised version of the paper and in the general response with the results of the revised 3DGraphLLM method on the ScanQA, SQA3D, and Scan2Cap datasets. As shown in Table 2 (and Table 1 in the general response), the use of the scene graph also has a positive impact on solving the Scene Captioning task while maintaining comparable performance on the Question Answering tasks, ScanQA and SQA3D.
>
> > The boost in performance by using the proposed scene graph technique is only around 1-2% (Table-4) in the setup when models uses masks from a detector like Mask3D (which represents the realistic setup). This boost comes at a cost of additional complexities of making the scene graphs and querying the LLMs with additional tokens (additional ~800 tokens per query).
>
> The primary contribution of our work is a method for creating a learnable representation of the scene graph for LLMs, which improves the performance of 3D Vision-Language tasks compared to methods that do not utilize semantic relationships between objects. When using ground-truth segmentation of the scene's point cloud, the performance gain from incorporating the scene graph reaches 5-8% (Table 4, Table 5 of the paper) for 3D Referred Object Grounding tasks. Note that in the revised version of the method, the performance boost due to the scene graph is 2-3% for LLAMA-8B-Instruct (Table 1 in the general response). Additionally, as shown in Table 1 in the general response, the use of the scene graph improves the performance of the Scene Captioning task (+3-4% CIDEr@0.5) while maintaining comparable performance for the Question Answering task. Finally, despite the use of additional tokens to describe the scene, adding the scene graph increases the inference time of the LLM in the 3D Referred Object Grounding task by only 7% (from 140 to 150 ms, see Figure 5 in the paper). For graph construction, we use the pre-trained VL-SAT model, which processes a scene with 9 objects in approximately 9 ms to construct a complete graph.
>
> > The paper claims to achieve SOTA results on ScanRefer and Multi3DRefer benchmarks in the introduction. These claims do not look correct -- as per Table-2, 3D Vista and M3DRef-CLIP outperforms the proposed model on ScanRefer. Besides several stronger baselines like BUTD-DETR [1] and PQ3D [2] outperform 3DGraphLLMs by large margins. Perhaps, the intention was to say that 3D GraphLLMs is SOTA LLM-based visual grounding model -- the claim in introduction may likely need to be revised appropriately. 3DGraphLLMs outperformM3DRef-CLIP baseline on Multi3DRefer benchmark. However, the baseline is trained on much lesser data compared to 3DGraphLLMs.
>
> The revised version of 3DGraphLLM demonstrates superior performance compared to expert models, including stronger ones such as BUTD-DETR  and PQ3D (see Table 2). Please, refer to Table 2 in the general response.
>
> [1] Huang, H., Chen, Y., Wang, Z., Huang, R., Xu, R., Wang, T., ... & Zhao, Z. (2024). Chat-scene: Bridging 3d scene and large language models with object identifiers. In The Thirty-eighth Annual Conference on Neural Information Processing Systems.

---

> > ### Comment · Reviewer_qz74 · 2024-11-25
> >
> > Thank you to the authors for their response. Below are my comments:
> >
> > - Comparison with same LLMs: thank you for conducting this experiment. This makes the comparison more fair. Compared to ChatScene, my understanding is that the performance improvement is 1.5% with a comparable LLM for the proposed method.
> >
> > - Thank you for reporting results on additional datasets as well.
> >
> > - About performance boost with additional 3d tokens: thank you for explaining it. In my opinion, the key experiment to take into consideration are the ones with detected boxes instead of GT boxes. I understand that the performance improvement there is in the order of 2-3%. I am not sure I agree that the statement: "VL-SAT builds object representation for 9 objects in 9ms" to be representative of the additional runtime, as ScanNet scenes in general have an order of 50-100 objects per scene (see ScanNet200 annotations) and detectors like Mask3D produce 150 object proposals.
> >
> > - Revised performance of 3D GraphLLMs: The earlier reported performance, at the time of submission, was 49% on ScanRefer and the revised performance is 60%. I believe this increase is mainly due to shifting from Chat3D-v2 as base LLM to ChatScene (and using LLAMA-3B). I would advise the authors to make this very clear in the common response -- as this was not obvious to me. While I understand that the underlying base LLM usage is orthogonal to the contributions of this work, I am not supportive of such major changes during the discussion / rebuttal time as this essentially disregards the submission deadline and uses extra time to alter major results -- instead of simply clarifying the concerns. A consequence of this change is also that the numbers in the paper are currently inconsistent in my understanding: Table-2 uses ChatScene based numbers while the ablations of Table-3 and Table-4 seems to be using Chat3D-v2 based numbers.
> >
> > After carefully reviewing the work and author's response I am not supportive of this paper's publication at this time for two main reasons: a) This is subjective opinion (and I am happy to change based on discussion with other reviewers), but I think that the performance improvement of this method is not significant compared to additional complexities it adds b) I am not supportive of the major changes in main results of the paper after the submission deadline and the subsequent inconsistencies in the results of the paper (between main results and the ablations). If other reviewers and AC are happy to look past this, I would be happy to drop this concern.
> >
> > I want to again thank the authors for their hard work and response. I think a proper revision of this work can help make this work stronger.

---

> > > ### Author Response · Authors · 2024-11-27
> > >
> > > Thank you for the feedback on the revised version of the paper. We would like to answer to your comments:
> > >
> > > a) When considering LLM Vicuna 1.5 used in other works, the performance improvement observed is +3% F1@0.25 and +3% F1@0.5 for Multi3DRefer, +4.1 CIDEr@0.5 for Scan2Cap, and +1.5% Acc@0.25 and +1.1% Acc@0.5 for ScanRefer. Meanwhile, for the ScanRefer dataset, using the 3DGraphLLM graph-based scene representation provides a more significant improvement for LLM LLAMA3-8B-Instruct. Specifically, the improvement is +2.6% for Acc@0.5 compared to the baseline Chat-Scene method using LLAMA3-8B-Instruct.
> > > It is worth noting that for 3D Referred Object Grounding and Scan Captioning, such an improvement is considered quite significant. For instance, the authors of the UniT3D [1] method propose an approach that achieves a 1.1% increase in Acc@0.5 on ScanRefer compared to BUTD-DETR and a 4.4% increase in CIDEr@0.5 on Scan2Cap compared to D3Net. Moreover, their proposed pretraining method yields a 2.6% improvement in Acc@0.5 on ScanRefer and a 3.6% improvement in CIDEr@0.5 on Scan2Cap.
> > >
> > > b) The changes we made to the paper do not affect our main contributions: the method for creating a learnable graph-based scene representation, which improves performance on 3D vision-language tasks by leveraging semantic relationships between objects, as well as the algorithm for generating a flat sequence of the scene graph, representing an object as a subgraph of its nearest neighbors. The changes made have improved the convergence of the 3DGraphLLM method and addressed the main concerns raised by the reviewers while keeping the core contributions unchanged.
> > > In our view, our main contribution lies in improving the performance of existing object-centric 3D-LLMs by incorporating a graph-based scene representation with semantic edges. Therefore, both parts of results remain consistent in the sense that the quality improvement provided by the graph-based scene representation remains unchanged, whether Chat3Dv2 or Chat-Scene is used as the base 3D object-centric LLM.
> > >
> > > [1] Chen, Z., Hu, R., Chen, X., Nießner, M., & Chang, A. X. (2023). Unit3d: A unified transformer for 3d dense captioning and visual grounding. In Proceedings of the IEEE/CVF international conference on computer vision (pp. 18109-18119).

---

> > > > ### Comment · Reviewer_qz74 · 2024-11-28
> > > >
> > > > Thank you to the authors for their response.
> > > >
> > > > I am confused about: "Chat-Scene method using LLAMA3-8B-Instruct". My understanding is that ChatScene uses Vicuna1.5-7B as you pointed out in your first response. I do not see additional numbers of ChatScene with LLAMA3-8B.
> > > >
> > > > I acknowledge the response of the authors about the changes made in the paper. As I explained in my prior response, they are major changes that lead to a change of about 10% performance after the submission deadline in the main results of the paper; and the ablations tables are inconsistent due to this change. The authors mention that ChatScene is closest method to their work in their original response, a paper which was not even cited in the submission version (in my understanding) which reflects the severity of the changes that took place. The reviewers should not be expected to go through all the results and ablations again following a major change and I believe it also discredits the sanctity of the paper submission deadline. I do not discredit the effort the authors put in getting the final results that they report now, and in engaging with reviewers in a discussion -- I am not supportive of major changes in the quantitaive results after the submission deadline and I would be unable to go through the paper and results again (if the authors fix the inconsistencies) to be confident about my vote of acceptance. Hence, I am sorry but I am not supportive of accepting this paper at this time.

---

> > > > > ### Author Response · Authors · 2024-11-30
> > > > >
> > > > > Thank you for the feedback. We would like to clarify that by "Chat-Scene method using LLAMA3-8B-Instruct," we mean our reproduction of the Chat-Scene method for 3D scene representation, where we replace the original Vicuna1.5-7B with LLAMA3-8B-Instruct while keeping the method of representing 3D scene features the same as proposed by the authors of Chat-Scene. We conducted this experiment for a more detailed analysis of the effect of applying our proposed approach, 3DGraphLLM, compared to Chat-Scene, using the same base LLM model, LLAMA3-8B-Instruct.

---

### Author Response · Authors · 2024-11-23
**Rebuttal Revision of 3DGraphLLM**

We thank the reviewers for their detailed and thoughtful feedback on our work. We are pleased that the reviewers acknowledged the motivation behind the proposed idea of incorporating semantic edges of the scene graph to enhance the quality of LLM responses in the 3D referred object grounding task. We also appreciate the reviewers' comments on the quality of the paper's writing and the ablation experiments conducted.
The main concerns raised by the reviewers focused on two points:
 - The ability of 3DGraphLLM to leverage the pre-trained LLM and the scene graph for solving other 3D vision-language tasks (raised by r. qz74 and r. cnDL).
- The design choice for filtering duplicate objects in the graph based on the minimum distance between nearest neighbors (raised by r. aDrA and r. cnDL).

We have addressed these concerns by modifying the method for generating the sequence describing an object, adding a 2D object token next to the object identifier token. This modification, inspired by the Chat-Scene method [1], improved the stability of the 3DGraphLLM's convergence without altering the core contribution of this work: developing a method for creating a learnable representation of the 3D semantic scene graph and an algorithm for generating a flat sequence from the graph based on the k-nearest neighbors of the object.
Additionally, we conducted further experiments with the Non-Maximum Suppression filter and an ablation study on the proposed training pipeline using GT pre-training.

We have revised the text of our paper based on the reviewers' comments.

Below, we provide the tables referenced in our responses to the reviewers. C denotes the CIDEr metric.

**Table 1**. Ablation study on semantic edges role and training pipeline.
| Methods | Pre-train | Number of edges | ScanRefer A@0.5 | Multi3DRefer F1@0.5 | ScanQA C | ScanQA B-4 | Sqa3D EM | Scan2Cap C@0.5 | Scan2Cap B-4@0.5 |
|-------------|------------|--------------|----------------------|--------------------------|---------------|----------------|---------------|---------------------|-----------------------|
| 3DGraphLLM-0 Vicuna1.5 | ✗ | 0 | 50.2 | 52.4 | 87.7 | 14.3 | 54.6 | 77.1 | 36.3 |
| 3DGraphLLM-2 Vicuna1.5 | ✗ | 2 | 50.1 | 52.7 | 92.2 | 15.5 | 54.7 | 80.4 | 36.9 |
| 3DGraphLLM-2 Vicuna1.5 | ✓ | 2 | 51.3 | 55.4 | 87.6 | 12.1 | 53.1 | 81.2 | 36.3 |
| 3DGraphLLM-0 LLAMA3-8B | ✗ | 0 | 52.0 | 55.1 | 84.0 | 15.8 | 53.8 | 80.0 | 37.5 |
| 3DGraphLLM-2 LLAMA3-8B | ✗ | 2 | 54.3 | 57.3 | 87.4 | 14.9 | 54.5 | 85.6 | 39.6 |
| 3DGraphLLM-2 LLAMA3-8B | ✓ | 2 | 54.6 | 58.2 | 83.1 | 12.5 | 55.2 | 82.9 | 37.8 |


**Table 2**.  Performance comparison of 3DGraphLLM with state-of-the-art approaches for 3D vision-language tasks. The ellipsis in the table indicates omitted baselines that were given in the original version of the paper.
|  |   | ScanRefer | Multi3DRefer |  |  | ScanQA | | Sqa3D | Scan2Cap |  |
| ---------- | --------- | --------- | ----------- | -------- | -------- | --------- | -------- | --------- | --------- | ---------- |
|    | Methods | A@0.25 | A@0.5 | F1@0.25 | F1@0.5 | C | B4 | EM | C@0.5 | B-4@0.5 |
| Expert models | ... | - | - | - | - | - | - | - | - | - |
|  | M3DRef-CLIP | 51.9 | 44.7 | 42.8 | 38.4 | - | - | - | - | - |
|  | Scan2Cap | - | - | - | - | - | - | - | 35.2 | 22.4 |
|  | ScanQA | -  | - | - | - | 64.9 | 10.1| - | -| - |
|  | Sqa3D | - | - | - | - | - | - | 47.2 | - | - |
|  | 3D-VisTA | 50.6 | 45.8 | - | - | 72.9 | 13.1 | 48.5 | 66.9 | 34.0 |
|  | BUTD-DETR | 52.2 | 39.8 | - | - | - | - | - | - | - |
|  | PQ3D | - | 51.2 | - | 50.1 | 87.8 | - | 47.1 | 80.3 | 36.0 |
| LLM-based models | ... | - | - | - | - | - | - | - | - | - |
|  | Chat-3D v2 | 35.9 | 30.4 | - | - | 77.1 | 7.3 | - | - | - |
|  | Scene-LLM | - | - | - | - | 80.0 | 12.0 | 54.2 | - | - |
|  | LL3DA | - | - | - | - | 76.8 | 13.5 | - | 65.2 | 36.8 |
|  | Grounded 3D-LLM | 47.9 | 44.1 | 45.2 | 40.6 | 72.7 | 13.4 | - | 70.6 | 35.5 |
|  | Chat-Scene | 55.5 | 50.2 | 57.1 | 52.4 | 87.7 | 14.3 | 54.6 | 77.1 | 36.3 |
|  | 3DGraphLLM Vicuna-1.5 | 57.0 | 51.3 | 60.1 | 55.4 | 87.6 | 12.1 | 53.1 | 81.2 | 36.3 |
|  | 3DGraphLLM LLAMA3-8B | 60.2 | 54.6 | 63.0 | 58.2 | 83.1 | 12.5 | 55.2 | 82.9 | 37.8 |

**Table 3**. Ablation study on semantic edges role depending on quality of instance segmentation.
| Methods | Instance segmentation | Number of edges | Minimal distance (cm) | ScanRefer A@0.25 | ScanRefer A@0.5 |
| ---------- | ------------------------- | ------------------- | ------------------------- | ----------------------- | ---------------------- |
| 3DGraphLLM-0 | Mask3D | 0 | - | 46.0 | 34.2 |
| 3DGraphLLM-2 | Mask3D | 2 | 0 | 47.3 (+ 1.3%) | 35.6 (+ 1.4%) |
| 3DGraphLLM-2 | Mask3D | 2 | 1 | 48.0 (+ 2.0%) | 36.2 (+ 2.0%) |
| 3DGraphLLM-2 | Mask3D (+ NMS) | 2 | 1 | 48.1 (+ 2.1%) | 36.5 (+ 2.3%) |

[1] Huang, H., et al. (2024). Chat-scene: Bridging 3d scene and large language models with object identifiers. In The 38 Annual Conference on Neural Information Processing Systems.

---

> ### Comment · Reviewer_cnDL · 2024-11-25
>
> Thanks for the updated results!
>
> I notice that in Table 1 of this response, some numbers are drastically different from those in Table 3 in the first version of the paper, across the board: ScanQA CIDEr metrics go up from ~67 to 83, ScanQA B-4 go from 7.8 to 12-15. Similar jumps also exist in all rows for Sqa3D and Scan2Cap.
>
> The same thing happens between Table 2 in this response and Table 2 in the first version of the paper, where the baselines have similar numbers but 3DGraphLLM's numbers go up by 7-15 points.
>
> Seems Table 2 and Table 3 in this response also have inconsistent results. Why 3DGraphLLM in Table 2 is better than 3DGraphLLM in Table 3 by >10 percent?
>
> Can authors explain what causes this huge difference in the results?

---

> > ### Author Response · Authors · 2024-11-27
> >
> > Thank you for your response. The performance improvements in Tables 2 and 3 in the updated version of the paper are attributed to using Chat-Scene as the baseline object-centric scene representation for LLM, which includes an additional 2D object token derived from multi-view images of the object. The inclusion of the 2D object token also improved the performance of 3DGraphLLM. Table 3 in the response (Table 4 in the paper) refers to the version of 3DGraphLLM without the image token.
> > In Table 1, "pre-train" indicates the addition of pre-training on graphs generated using GT instance segmentation of ScanNet scenes, followed by fine-tuning on graphs generated using Mask3D instance segmentation as object proposals for ScanNet scenes.

---

> > > ### Comment · Reviewer_cnDL · 2024-11-28
> > >
> > > Can authors clarify whether including the masked image features in 3DGraphLLM still constitutes a fair comparison to the baselines? I suggest the authors clearly note which baselines use 3D input only and which use 2D and 3D input.

---

> > > > ### Author Response · Authors · 2024-11-30
> > > >
> > > > We believe that, yes, using image features still constitutes a fair comparison with baseline methods. In fact, leveraging dense 2D features extracted from RGBD sequences to improve the quality of solving 3D vision-language tasks is a common practice (see the table below). However, different authors employ various approaches to effectively map 2D observations obtained from different views to 3D objects features. In the case of 3DGraphLLM, we follow the method proposed by the authors of Chat-Scene. It is worth noting that, in a real-world scenario, if we assume an RGB-D scan of a scene is provided, the sequence of RGB images used to construct it is usually available as well, meaning we can utilize this data. We include information in the table below about which methods also use features extracted from 2D multi-view images as model input in the comparison with baseline methods.
> > > >
> > > > |  |   |   | ScanRefer | Multi3DRefer |   |  | ScanQA |          | Sqa3D | Scan2Cap |  |
> > > > | -------------------- | ---- | --------------------- | ------------- | ---------------- | -------- | -------- | ---------- | -------- | --------- | ------------ | -------- |
> > > > |    | Methods | 2D features  | A@0.25↑ | A@0.5↑ | F1@0.25↑ | F1@0.5↑ | C↑ | B4↑ | EM↑ | C@0.5↑ | B-4@0.5↑ |
> > > > | Expert models | ScanRefer | ✓| 37.3 | 24.3 | - | - | - | - | - | - | - |
> > > > |  | MVT | ✓ | 40.8 | 33.3 | - | - | - | - | - | - | - |
> > > > |  | 3DVG-Trans | ✗ | 45.9 | 34.5 | - | - | - | - | - | - | - |
> > > > |  | 3DVG-Trans | ✓ | 47.6 | 34.7 | - | - | - | - | - | - | - |
> > > > |  | ViL3DRel | ✗ | 47.9 | 37.7 | - | - | - | - | - | - | - |
> > > > |  | M3DRef-CLIP | ✓ | 51.9 | 44.7 | 42.8 | 38.4 | - | - | - | - | - |
> > > > |  | Scan2Cap | ✓ | - | - | - | - | - | - | - | 35.2 | 22.4 |
> > > > |  | ScanQA | ✓ | -  | - | - | - | 64.9 | 10.1| - | -| - |
> > > > |  | Sqa3D | ✗ | - | - | - | - | - | - | 47.2 | - | - |
> > > > |  | 3D-VisTA | ✗ | 50.6 | 45.8 | - | - | 72.9 | 13.1 | 48.5 | 66.9 | 34.0 |
> > > > |  | BUTD-DETR | ✗ | 52.2 | 39.8 | - | - | - | - | - | - | - |
> > > > |  | PQ3D | ✓ | - | 51.2 | - | 50.1 | 87.8 | - | 47.1 | 80.3 | 36.0 |
> > > > | LLM-based models | ZSVG3D | ✓ |  36.4  | 32.7 | - | - | - | - | - | - | - |
> > > > |  | 3D-LLM(Flamingo) | ✓ | 21.2  | - | -  | - | 59.2 | 7.2 | - | - | - |
> > > > |  | 3D-LLM(BLIP2-flant5) | ✓ | 30.3  | - | - | - | 69.4  | 12.0  | - | - | - |
> > > > |  | Chat-3D v2 | ✗ | 35.9 | 30.4 | - | - | 77.1 | 7.3 | - | - | - |
> > > > |  | Scene-LLM | ✓ | - | - | - | - | 80.0 | 12.0 | 54.2 | - | - |
> > > > |  | LL3DA | ✗ | - | - | - | - | 76.8 | 13.5 | - | 65.2 | 36.8 |
> > > > |  | Grounded 3D-LLM | ✗ | 47.9 | 44.1 | 45.2 | 40.6 | 72.7 | 13.4 | - | 70.6 | 35.5 |
> > > > |  | Chat-Scene | ✓ | 55.5 | 50.2 | 57.1 | 52.4 | 87.7 | 14.3 | 54.6 | 77.1 | 36.3 |
> > > > |  | 3DGraphLLM Vicuna-1.5 | ✓ | 57.0 | 51.3 | 60.1 | 55.4 | 87.6 | 12.1 | 53.1 | 81.2 | 36.3 |
> > > > |  | 3DGraphLLM LLAMA3-8B | ✓ | 60.2 | 54.6 | 63.0 | 58.2 | 83.1 | 12.5 | 55.2 | 82.9 | 37.8 |

---

### Author Response · Authors · 2024-11-30

We would like to thank the reviewers for their feedback and for raising their scores. We are pleased that our responses addressed their initial concerns. We would like to emphasize that most of the changes made based on the reviewers' feedback have already been incorporated into the revised version of the manuscript. The remaining results obtained during the discussion will be included in the final version of the paper.

---

### Meta-Review · Area_Chair_h58d · 2024-12-24

**Metareview:**

This paper proposes 3DGraphLLM to combine semantic graphs and LLMs for 3D referred object grounding in scenes. Initially, the reviewers raised concerns about the experimental comparisons and the marginal performance gains. The authors provided a strong rebuttal addressing many of these issues. However, several reviewers noted that the experiments in the revised submission differ significantly from the original, and the paper's writing requires substantial improvement.

While the paper demonstrates notable merit, it is not recommended for acceptance in its current form. The authors are encouraged to incorporate the reviewers' feedback when revising the paper for submission to other venues.

**Additional Comments On Reviewer Discussion:**

The rebuttal does not address the concerns regarding novelty and the quality of the paper's writing very well.

---

### Decision · Program_Chairs · 2025-01-22

Reject